# Transparent Networks for Multivariate Time Series

## Abstract

Transparent models, which are machine learning models that produce inherently interpretable predictions, are receiving significant attention in high-stakes domains. However, despite much real-world data being collected as time series, there is a lack of studies on transparent time series models. To address this gap, we propose a novel transparent neural network model for time series called Generalized Additive Time Series Model (GATSM). GATSM consists of two parts: 1) independent feature networks to learn feature representations, and 2) a transparent temporal module to learn temporal patterns across different time steps using the feature representations. This structure allows GATSM to effectively capture temporal patterns and handle dynamic-length time series while preserving transparency. Empirical experiments show that GATSM significantly outperforms existing generalized additive models and achieves comparable performance to black-box time series models, such as recurrent neural networks and Transformer. In addition, we demonstrate that GATSM finds interesting patterns in time series. The source code is available at https://anonymous.4open.science/r/GATSM-78F4/.

## 1 Introduction

Artificial neural networks excel at learning complex representations and demonstrate remarkable predictive performance across various fields. However, their complexity makes interpreting the decision-making processes of neural network models challenging. Consequently, post-hoc explainable artificial intelligence (XAI) methods, which explain the predictions of trained black-box models, have been widely studied in recent years [1, 2, 3, 4]. XAI methods are generally effective at providing humans with understandable explanations of model predictions. However, they may produce incorrect and unfaithful explanations of the underlying black-box model and cannot provide actual contributions of input features to model predictions [5, 6]. Therefore, their applicability to high-stakes domains-such as healthcare and fraud detection, where faithfulness to the underlying model and actual contributions of features are important-is limited.

Due to these limitations, transparent (i.e., inherently interpretable) models are attracting attention as alternatives to XAI in high-stakes domains [7, 8, 9]. Modern transparent models typically adhere to the *generalized additive model* (GAM) framework [10]. A GAM consists of independent functions, each corresponding to an input feature, and makes predictions as a linear combination of these functions (e.g., the sum of all functions). Therefore, each function reflects the contribution of its respective feature. For this reason, interpreting GAMs is straightforward, making them widely used in various fields, such as healthcare [11, 12], survival analysis [13], and model bias discovery [7, 14, 15]. However, despite much real-world data being collected as time series, research on GAMs for time series remains scarce. Consequently, the applicability of GAMs in real-world scenarios is still limited.

To overcome this limitation, we propose a novel transparent model for multivariate time series called Generalized Additive Time Series Model (GATSM). GATSM consists of independent feature networks to learn feature representations and a transparent temporal module to learn temporal patterns.

Submitted to 38th Conference on Neural Information Processing Systems (NeurIPS 2024). Do not distribute.

Since employing distinct networks across different time steps requires a massive amount of learnable parameters, the feature networks in GATSM share the weights across all time steps, while the temporal module independently learns temporal patterns. GATSM then generates final predictions by integrating the feature representations with the temporal information from the temporal module. This strategy allows GATSM to effectively capture temporal patterns and handle dynamic-length time series while preserving transparency. Additionally, this approach facilitates the separate extraction of time-independent feature contributions, the importance of individual time steps, and time-dependent feature contributions through the feature functions, temporal module, and final prediction. To demonstrate the effectiveness of GATSM, we conducted empirical experiments on various time series datasets. The experimental results show that GATSM significantly outperforms existing GAMs and achieves comparable performances to black-box time series models, such as recurrent neural networks and Transformer [16]. In addition, we provide visualizations of GATSM's predictions to demonstrate that GATSM finds interesting patterns in time series.

## 2 Related Works

Various XAI studies have been conducted over the past decade [7, 8, 9, 17, 18]; however, they are less relevant to the transparent model that is the subject of this study. Therefore, we refer readers to [19, 20] for more detailed information on recent XAI research. In this section, we review existing transparent models closely related to our GATSM and discuss their limitations.

Table 1: Advantages of GATSM.

|  | Time series input | Temporal pattern | Dynamic time series |
| --- | --- | --- | --- |
| existing GAMs |  |  |  |
| NATM | ✓ |  |  |
| GATSM (our) | ✓ | ✓ | ✓ |

The simple linear model is designed to fit the conditional expectation $g\left(\mathbb{E}\left(y \mid \mathbf{x}\right)\right) = \sum_{i=1}^{M} x_i w_i$, where $g(\cdot)$ is a link function, $M$ indicates the number of input features, $y$ is the target value for the given input features $\mathbf{x} \in \mathbb{R}^M$, and $w_i \in \mathbb{R}$ is the learnable weight for $x_i$. This model captures only linear relationships between the target $y$ and the inputs $\mathbf{x}$. To address this limitation, GAM [10] extends the simple linear model to the generalized form as follows:

$$g\left(\mathbb{E}\left(y \mid \mathbf{x}\right)\right) = \sum_{i=1}^{M} f_i\left(x_i\right), \tag{1}$$

where each $f_i(\cdot)$ is a function that models the effect of a single feature, referred as a feature function. Typically, $f_i(\cdot)$ becomes a non-linear function such as a decision tree or neural network to capture non-linear relationships.

Originally, GAMs were fitted via the backfitting algorithm using smooth splines [10, 21]. Later, Yin Lou et al. [22] and Harsha Nori et al. [23] have proposed boosted decision tree-based GAMs, which use boosted decision trees as feature functions. Spline- and tree-based GAMs have less flexibility and scalability. Thus, extending them to transfer or multi-task learning is challenging. To overcome this problem, various neural network-based GAMs have been proposed in recent years. Potts [24] introduced generalized additive neural network, which employs 2-layer neural networks as feature functions. Similarly, Rishabh Agarwal et al. [7] proposed neural additive model (NAM) that employs multi-layer neural networks. To improve the scalability of NAM, Chun-Hao Chang et al. [8] and Filip Radenovic et al. [9] proposed the neural oblivious tree-based GAM and the basis network-based GAM, respectively. Xu et al. [25] introduced a sparse version of NAM using the group LASSO. One disadvantage of GAMs is their limited predictive power, which stems from the fact that they only learn first-order feature interactions-i.e., relationships between the target value and individual features. To address this, various studies have been conducted to enhance the predictive powers of GAMs by incorporating higher-order feature interactions, while still maintaining transparency. GA$^2$M [26] simply takes pairwise features as inputs to learn pairwise interactions. GAMI-Net [27], a neural network-based GAM, consists of networks for main effects (i.e., first-order interactions) and pairwise interactions. To enhance the interpretability of GAMI-Net, the sparsity and heredity constraints are added, and trivial features are pruned in the training process. Sparse interaction additive network [28]

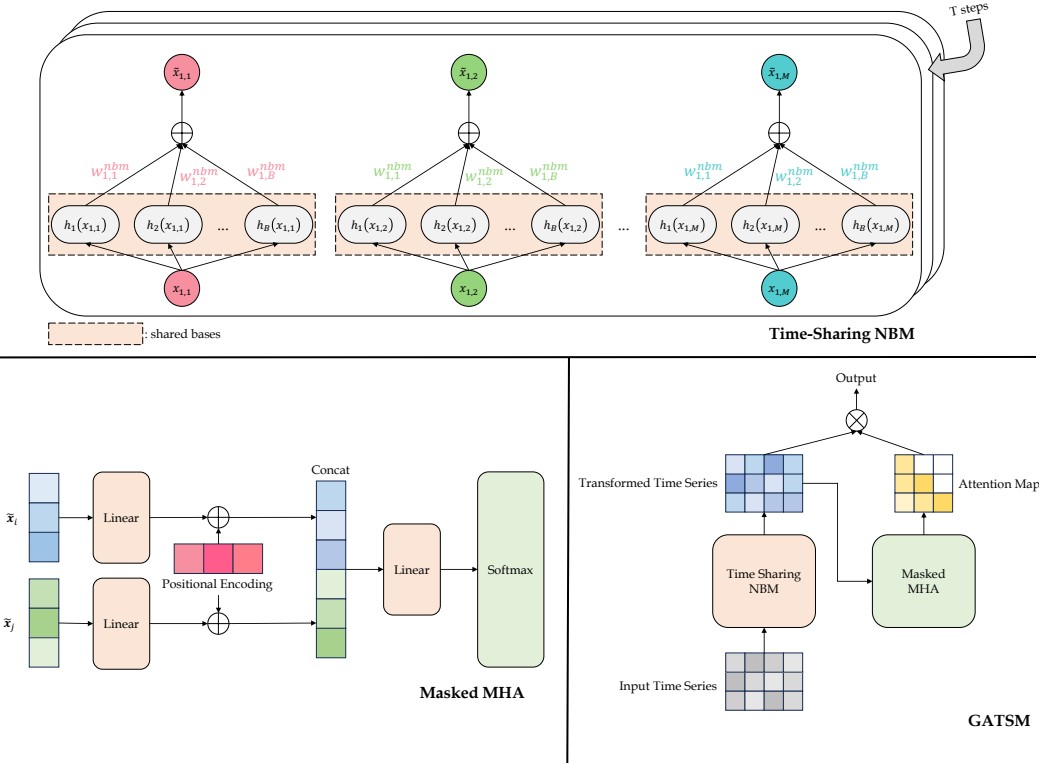

Figure 1: Architecture of GATSM.

is a 3-phase method for exploiting higher-order interactions. Initially, a black-box neural network is trained; subsequently, the top-$k$ important features are identified using explainable feature attribution methods like LIME [1] and SHAP [2], and finally, NAM is trained with these extracted features. Dubey et al. [29] introduced scalable polynomial additive model, an end-to-end model that learns higher-order interactions via polynomials. Similarly, Kim et al. [15] proposed higher-order NAM that utilizes the feature crossing technique to capture higher-order interactions. Despite their capabilities, the aforementioned GAMs cannot process time series data, which limits their applicability in real-world scenarios. Recently, neural additive time series Model (NATM) [30], a time-series adaptation of NAM, has been proposed. However, NATM handles each time step independently with separate feature networks. This approach cannot capture effective temporal patterns and only takes fixed-length time series as input. Our GATSM not only captures temporal patterns but also handles dynamic-length time series. Table 1 shows the advantages of our GATSM compared to existing GAMs.

## 3 Problem Statement

We tackle the problem of the existing GAMs on time series. Equation (1) outlines the GAM framework for tabular data, which fails to capture the interactions between current and previous observations in time series. A straightforward method to extend GAM to time series, adopted in NATM, is applying distinct feature functions to each time step and summing them to produce predictions:

$$g\left(\mathbb{E}\left(y_t \mid \mathbf{X}_{:t}\right)\right) = \sum_{i=1}^{t} \sum_{j=1}^{M} f_{i,j}\left(x_{i,j}\right), \tag{2}$$

where $\mathbf{X} \in \mathbb{R}^{T \times M}$ is a time series with $T$ time steps and $M$ features, and $t$ is the current time step. This method can handle time series data as input but fails to capture effective temporal patterns because the function $f_{i,j}\left(\cdot\right)$ still does not interact with previous time steps. To overcome this problem,

we suggest a new form of GAM for time series defined as follows:

$$g\left(\mathbb{E}\left(y_t \mid \mathbf{X}_{:t}\right)\right) = \sum_{i=1}^{t} \sum_{j=1}^{M} f_{i,j}\left(x_{i,j}, \mathbf{X}_{:t}\right). \tag{3}$$

**Definition 3.1** *GAMs for time series, which capture temporal patterns hold the form of Equation 3.*

In Equation (3), the function $f\left(\cdot, \cdot\right)$ can capture interactions between current and previous time steps. Therefore, GAMs adhering to Definition 3.1 are capable of capturing temporal patterns. However, implementing such a model while maintaining transparency poses challenges. In the following section, we will describe our approach to implementing a GAM that holds Definition 3.1. To the best of our knowledge, no existing literature addresses Definition 3.1.

# 4 Our Method: Generalized Additive Time Series Model

## 4.1 Architecture

Figure 1 shows the overall architecture of GATSM. Our model has two modules: 1) feature networks, called time-sharing neural basis model, for learning feature representations, and 2) masked multi-head attention for learning temporal patterns.

**Time-Sharing NBM:** Assume a time series with $T$ time steps and $M$ features. Applying GAMs to this time series necessitates $T \times M$ feature functions, which becomes problematic when dealing with large $T$ or $M$ due to increased model size. This limits the applicability of GAMs to real-world datasets. To overcome this problem, we extend neural basis model (NBM) [9] to time series as:

$$\tilde{x}_{i,j} = f_j\left(x_{i,j}\right) = \sum_{k=1}^{B} h_k\left(x_{i,j}\right) w_{j,k}^{nbm}. \tag{4}$$

We refer to this extended version of NBM as time-sharing NBM. Time-sharing NBM has $B$ basis functions, with each basis $h_k(\cdot)$ taking a feature $x_{i,j}$ as input. The feature-specific weight $w_{j,k}^{nbm}$ then projects the basis to the transformed feature $\tilde{x}_{i,j}$. As depicted in Equation 4, the basis functions are shared across all features and time steps, drastically reducing the number of required feature functions $T \times M$ to $B$. We use $B = 100$ and implement $h_k(\cdot)$ using multi-layer perceptron (MLP).

**Masked MHA:** GATSM employs multi-head attention (MHA) to learn temporal patterns. Although the dot product attention [16] is popular, simple dot operation has low expressive power [31]. Therefore, we adopt the 2-layer attention mechanism proposed by [31] to GATSM. We first transform $\tilde{\mathbf{x}}_i = [\tilde{x}_{i,1}, \tilde{x}_{i,2}, \cdots, \tilde{x}_{i,M}] \in \mathbb{R}^M$ produced by Equation 4 as follows:

$$\mathbf{v}_i = \tilde{\mathbf{x}}_i^\intercal \mathbf{Z} + \mathbf{pe}_i, \tag{5}$$

where $\mathbf{Z} \in \mathbb{R}^{M \times D}$ is a learnable weight, $\mathbf{pe}_i = [pe_{i,1}, pe_{i,2}, \cdots, pe_{i,D}] \in \mathbb{R}^D$ is the positional encoding for $i$-th step, and $D$ indicates the hidden size. The positional encoding is defined as follows:

$$pe_{i,j} = \begin{cases} \sin\left(\frac{i}{10000^{2j/D}}\right) & \text{if } j \bmod 2 = 1, \\ \cos\left(\frac{i}{10000^{2j/D}}\right) & \text{otherwise.} \end{cases} \tag{6}$$

The positional encoding helps GATSM effectively capture temporal patterns. While learnable position embedding also works in GATSM, we recommend positional encoding because position embedding requires knowledge of the maximum number of time steps, which is often unknown in real-world settings. After computing $\mathbf{v}_i$, we calculate the attention scores as follows:

$$e_{k,i,j} = \sigma\left([\mathbf{v}_i \mid \mathbf{v}_j]^\intercal \mathbf{w}_k^{attn}\right) m_{i,j}, \tag{7}$$

$$a_{k,i,j} = \frac{\exp\left(e_{k,i,j}\right)}{\sum_{t=1}^{T} \exp\left(e_{k,i,t}\right)}, \tag{8}$$

where $k$ is attention head index, $\sigma\left(\cdot\right)$ is an activation function, $\mathbf{w}_k^{attn} \in \mathbb{R}^{2D}$, and $m_{i,j} \in \mathbb{R}$ is the mask value used to block future information. The time mask is defined as follows:

$$m_{i,j} = \begin{cases} 1 & \text{if } i \leq j, \\ -\infty & \text{otherwise.} \end{cases} \tag{9}$$

136 **Inference:** The prediction of GATSM is produced by combining the transformed features from
137 time-sharing NBM with the attention scores from masked MHA.

$$\hat{y}_t = \sum_{k=1}^{K} \mathbf{a}_{k,t}^{\intercal} \tilde{\mathbf{X}} \mathbf{w}_k^{out}, \tag{10}$$

138 where $K$ is the number of attention heads, $\mathbf{a}_{k,t} = [a_{k,i,1}, a_{k,i,2}, \cdots, a_{k,i,T}] \in \mathbb{R}^T$ is the attention
139 map in Equation 8, $\tilde{\mathbf{X}} = [\tilde{\mathbf{x}}_1, \tilde{\mathbf{x}}_2, \cdots, \tilde{\mathbf{x}}_T] \in \mathbb{R}^{T \times M}$ is the transformed features in Equation 4, and
140 $\mathbf{w}_k^{out} \in \mathbb{R}^M$ is the learnable output weight.

141 **Interpretability:** We can rewrite Equation 10 as the following scalar form:

$$\sum_{k=1}^{K} \mathbf{a}_{k,t}^{\intercal} \tilde{\mathbf{X}} \mathbf{w}_k^{out} = \sum_{u=1}^{t} \sum_{m=1}^{M} \sum_{k=1}^{K} \sum_{b=1}^{B} a_{k,t,u} h_b \left( x_{t,m} \right) w_{m,b}^{nbm} w_{k,m}^{out}$$

$$= \sum_{u=1}^{t} \sum_{m=1}^{M} f_{u,m} \left( x_{u,m}, \mathbf{X}_{:t} \right) \tag{11}$$

142 Equation 11 shows that GATSM satisfying Definition 3.1. We can derive three types of interpretations
143 from GATSM: 1) $a_{k,t,u}$ indicates the importance of time step $u$ at time step $t$, 2) $h_b \left( x_{t,m} \right) w_{m,b}^{nbm} w_{k,m}^{out}$
144 represents the time-independent contribution of feature $m$, and 3) $a_{k,t,u} h_b \left( x_{t,m} \right) w_{m,b}^{nbm} w_{k,m}^{out}$ repre-
145 sents the time-dependent contribution of feature $m$ at time step $t$.

# 5 Experiments

## 5.1 Experimental Setup

148 **Datasets:** We conducted our experiments using eight publicly available real-world time series
149 datasets. From the Monash repository [32], we sourced three datasets: Energy, Rainfall, and
150 AirQuality. Another three datasets, Heartbeat, LSST, and NATOPS, were downloaded from the
151 UCR repository [33]. The remaining two datasets, Mortality and Sepsis, were downloaded from
152 the PhysioNet [34]. We perform ordinal encoding for categorical features and standardize features
153 to have zero-mean and unit-variance. For forecasting tasks, target value y is also standardized to
154 zero-mean and unit-variance. If the dataset contains missing values, we impute categorical features
155 with their modes and numerical features with their means. The dataset is split into a 60%/20%/20%
156 ratio for training, validation, and testing, respectively. Table 2 shows the statistics of the experimental
157 datasets. Further details of the experimental datasets can be found in Appendix B.

Table 2: Dataset statistics.

| Dataset | Task | Variable length | # of time series | Avg. length | # of features | # of classes |
|---|---|---|---|---|---|---|
| Energy | 1-step FCST | No | 137 | 24 | 24 | - |
| Rainfall | 1-step FCST | No | 160,267 | 24 | 3 | - |
| AirQuality | 1-step FCST | No | 16,966 | 24 | 9 | - |
| Heartbeat | Binary | No | 409 | 405 | 61 | 2 |
| Mortality | Binary | Yes | 12,000 | 49.861 | 41 | 2 |
| Sepsis | Binary | Yes | 40,336 | 38.482 | 40 | 2 |
| LSST | Multi-class | No | 4,925 | 36 | 6 | 14 |
| NATOPS | Multi-class | No | 360 | 51 | 24 | 6 |

FCST: forecasting

158 **Baselines:** We compare our GATSM with 12 baselines, which can be categorized into four groups: 1)
159 Black-box tabular models include extreme gradient boosting (XGBoost) [35] and MLP. 2) Black-box
160 time series models include simple recurrent neural network (RNN), gated recurrent unit (GRU), long
161 short-term memory (LSTM), and Transformer [16]. 3) Transparent tabular models are simple linear
162 model (Linear), explainable boosting machine (EBM) [23], NAM [7], NodeGAM [8], and NBM [9].
163 4) NATM [30] is a transparent time series model.

164 **Implementation:** We implement XGBoost and EBM models using the `xgboost` and `interpretml`
165 libraries, respectively. For NodeGAM, we employ the official implementation provided by its authors
166 [8]. The remaining models are developed using PyTorch [36]. All models undergo hyperparameter

Table 3: Predictive performance comparison of various models.

| Model Type | Model | Energy | Rainfall | AirQuality | Heartbeat | Mortality | Sepsis | LSST | NATOPS | Avg. Rank |
|---|---|---|---|---|---|---|---|---|---|---|
| Black-box Tabular Model | XGBoost | 0.094 (±0.137) | 0.002 (±0.002) | 0.532 (±0.019) | 0.679 (±0.094) | 0.707 (±0.015) | **0.816** (±0.007) | 0.424 (±0.012) | 0.200 (±0.049) | 8.500 (±4.000) |
| | MLP | 0.459 (±0.101) | 0.011 (±0.004) | 0.423 (±0.031) | 0.654 (±0.082) | 0.842 (±0.014) | 0.786 (±0.007) | 0.417 (±0.008) | 0.211 (±0.065) | 7.375 (±2.134) |
| Black-box Time Series Model | RNN | 0.320 (±0.122) | 0.068 (±0.020) | 0.644 (±0.032) | 0.661 (±0.078) | 0.581 (±0.040) | 0.782 (±0.009) | 0.422 (±0.029) | 0.592 (±0.110) | 7.750 (±2.712) |
| | GRU | 0.435 (±0.107) | 0.089 (±0.034) | 0.701 (±0.018) | 0.694 (±0.052) | 0.818 (±0.014) | 0.785 (±0.010) | 0.629 (±0.013) | 0.931 (±0.045) | 4.375 (±2.669) |
| | LSTM | 0.359 (±0.112) | 0.090 (±0.031) | 0.683 (±0.026) | 0.648 (±0.042) | 0.790 (±0.020) | 0.779 (±0.008) | 0.491 (±0.082) | 0.908 (±0.035) | 6.375 (±3.623) |
| | Transformer | 0.263 (±0.263) | **0.098** (±0.035) | **0.711** (±0.027) | 0.690 (±0.040) | 0.844 (±0.019) | 0.789 (±0.010) | **0.679** (±0.010) | **0.967** (±0.029) | 4.000 (±3.703) |
| Transparent Tabular Model | Linear | 0.482 (±0.112) | 0.004 (±0.001) | 0.241 (±0.019) | 0.637 (±0.070) | 0.838 (±0.017) | 0.723 (±0.011) | 0.311 (±0.010) | 0.206 (±0.045) | 10.125 (±3.871) |
| | EBM | -0.200 (±0.409) | 0.004 (±0.001) | 0.324 (±0.014) | 0.666 (±0.056) | 0.729 (±0.017) | 0.802 (±0.011) | 0.408 (±0.016) | 0.164 (±0.053) | 9.750 (±3.284) |
| | NAM | 0.363 (±0.218) | 0.006 (±0.002) | 0.300 (±0.013) | 0.645 (±0.026) | 0.853 (±0.014) | 0.800 (±0.006) | 0.400 (±0.011) | 0.242 (±0.040) | 7.875 (±3.643) |
| | NodeGAM | 0.398 (±0.195) | 0.006 (±0.002) | 0.380 (±0.032) | 0.681 (±0.046) | **0.854** (±0.013) | 0.802 (±0.007) | 0.400 (±0.028) | 0.247 (±0.012) | 6.375 (±3.623) |
| | NBM | 0.330 (±0.251) | 0.007 (±0.003) | 0.301 (±0.012) | 0.716 (±0.039) | 0.852 (±0.014) | 0.799 (±0.006) | 0.388 (±0.014) | 0.189 (±0.029) | 7.875 (±3.603) |
| Transparent Time Series Model | NATM | 0.304 (±0.122) | 0.038 (±0.011) | 0.548 (±0.028) | 0.724 (±0.043) | N/A | N/A | 0.452 (±0.010) | 0.878 (±0.058) | 5.667 (±2.582) |
| | GATSM (ours) | **0.493** (±0.173) | 0.073 (±0.027) | 0.583 (±0.026) | **0.843** (±0.025) | 0.853 (±0.015) | 0.797 (±0.007) | 0.570 (±0.024) | 0.956 (±0.027) | **3.125** (±1.808) |

tuning via Optuna [37]. The pytorch-based models are optimized with the Adam with decoupled weight decay (AdamW) [38] optimizer on an NVIDIA A100 GPU. Model training is halted if the validation loss does not decrease over 20 epochs. We use mean squared error for the forecasting tasks, and for classification tasks, we use cross-entropy loss. Further details of the model implementations and hyper-parameters are provided in Appendix C.

## 5.2 Comparison with baselines

Table 3 shows the predictive performances of the experimental models. We report mean scores and standard deviations over five different random seeds. For the forecasting datasets, we evaluate $R^2$ scores. For the binary classification datasets, we assess the area under the receiver operating characteristic curve (AUROC). For the multi-class classification datasets, we measure accuracy. We highlight the best-performing model in **bold** and underline the second-best model. Since the tabular models cannot handle time series, they only take $\mathbf{x}_t$ to produce $y_t$.

On the Energy and Heartbeat datasets, which are small in size, our GATSM demonstrates the best performance, indicating strong generalization ability. EBM, XGBoost, and Transformer struggle with overfitting on the Energy dataset. For the Mortality and Sepsis datasets, there is no significant performance difference between tabular and time series models, nor between black-box and transparent models. This suggests that these two healthcare datasets lack significant temporal patterns and feature interactions. It is likely that seasonal patterns are hard to detect in medical data, and the patient's current condition already encapsulates previous conditions, making historical data less crucial. Since these datasets contain variable-length time series, the performance of NATM, which can only handle fixed-length time series, is not available. On the Rainfall, AirQuality, LSST, and NATOPS datasets, the time series models significantly outperform the tabular models, indicating that these datasets contain important temporal patterns that tabular models cannot capture. Additionally, the black-box models outperform the transparent models, suggesting that these datasets have higher-order feature interactions that transparent models cannot capture. Nevertheless, GATSM is the best model within the transparent model group and performs comparably to Transformer. Overall, GATSM achieved the best average rank in the experiments, followed by the Transformer, indicating GATSM's superiority. Additional experiments on model throughput and an ablation study on the basis functions are presented in Appendix D.

Table 4: Ablation study on different feature functions.

| Feature Function | Energy | Rainfall | AirQuality | Heartbeat | Mortality | Sepsis | LSST | NATOPS |
|---|---|---|---|---|---|---|---|---|
| Linear | 0.283(±0.277) | 0.071(±0.024) | 0.563(±0.019) | 0.766(±0.024) | 0.832(±0.015) | 0.735(±0.012) | 0.398(±0.030) | **0.972**(±0.020) |
| NAM | 0.304(±0.229) | 0.068(±0.021) | 0.564(±0.019) | 0.838(±0.032) | 0.851(±0.013) | **0.801**(±0.005) | 0.553(±0.023) | 0.933(±0.039) |
| NBM | **0.493**(±0.173) | **0.073**(±0.027) | **0.583**(±0.026) | **0.843**(±0.025) | **0.853**(±0.015) | 0.797(±0.007) | **0.570**(±0.024) | 0.956(±0.027) |

Table 5: Ablation study on the temporal module.

| Temporal Module | Energy | Rainfall | AirQuality | Heartbeat | Mortality | Sepsis | LSST | NATOPS |
|---|---|---|---|---|---|---|---|---|
| Base | 0.452(±0.087) | 0.007(±0.002) | 0.299(±0.012) | 0.661(±0.043) | **0.854**(±0.013) | 0.798(±0.008) | 0.392(±0.006) | 0.192(±0.027) |
| Base + PE | 0.397(±0.054) | 0.007(±0.003) | 0.299(±0.012) | 0.681(±0.068) | 0.852(±0.013) | **0.799**(±0.007) | 0.385(±0.027) | 0.228(±0.029) |
| Base + MHA | 0.368(±0.230) | 0.048(±0.017) | 0.555(±0.020) | 0.821(±0.044) | 0.847(±0.020) | 0.779(±0.033) | **0.595**(±0.013) | 0.856(±0.059) |
| Base + PE + MHA | **0.493**(±0.173) | **0.073**(±0.027) | **0.583**(±0.026) | **0.843**(±0.025) | 0.853(±0.015) | 0.797(±0.007) | 0.570(±0.024) | **0.956**(±0.027) |

## 5.3 Ablation study

**Choice of feature function:** We evaluate the performance of GATSM by changing the feature functions using three models: Linear, NAM, and NBM. Table 4 presents the results of this experiment. The simple linear function performs poorly because it lacks the capability to capture non-linear relationships. In contrast, NAM, which can capture non-linearity, shows improved performance over the linear function. However, NBM stands out by achieving the best performance in six out of eight datasets. This indicates that the basis strategy of NBM is highly effective for time series data.

**Design of temporal module:** We evaluate the performance of GATSM by modifying the design of the temporal module. The results are presented in Table 5. GATSM without the temporal module (Base) fails to learn temporal patterns and shows poor performance in the experiment. GATSM with only positional encoding (Base + PE) also shows similar performance to the Base, indicating that positional encoding alone is insufficient for capturing effective temporal patterns. GATSM with only multi-head attention (Base + MHA) outperforms the previous two methods, demonstrating that the MHA mechanism is beneficial for capturing temporal patterns. Finally, our full GATSM (Base + PE + MHA) significantly outperforms the other methods, suggesting that the combination of PE and MHA creates a synergistic effect. Consistent with our previous findings in section 5.2, all four methods show similar performances on the Mortality and Sepsis datasets, which lack significant temporal patterns.

## 5.4 Interpretation

In this section, we visualize four interpretations of GATSM's predictions on the AirQuality dataset. In addition, interpretations for the Rainfall dataset can be found in Appendix E.

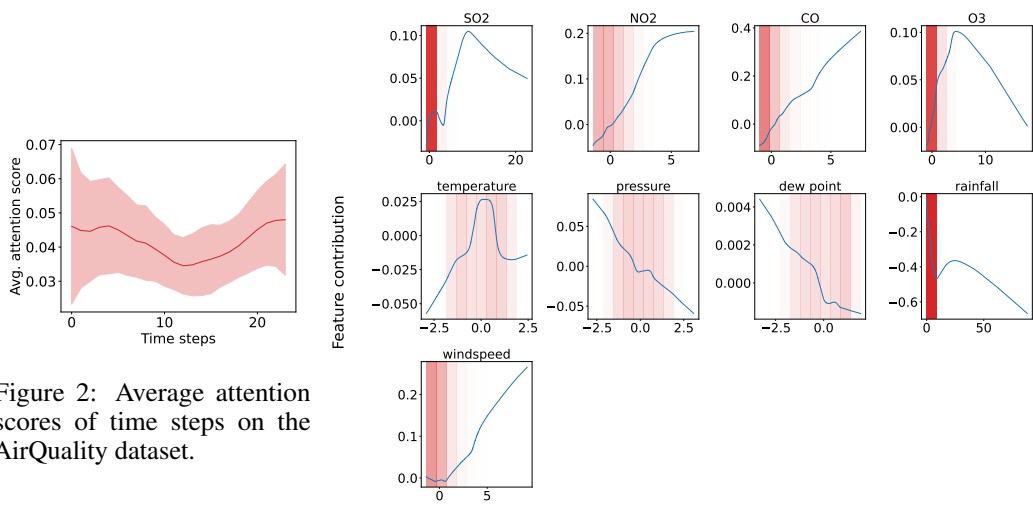

Figure 2: Average attention scores of time steps on the AirQuality dataset.

Figure 3: Global interpretations of features in the Air Quality dataset.

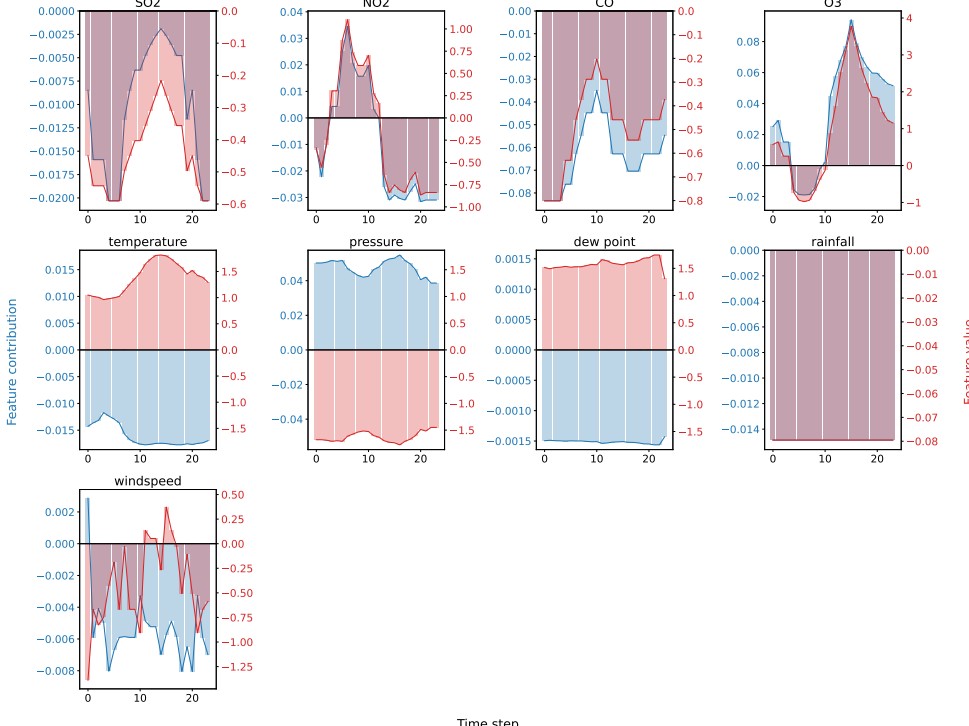

Figure 4: Local time-independent feature contributions.

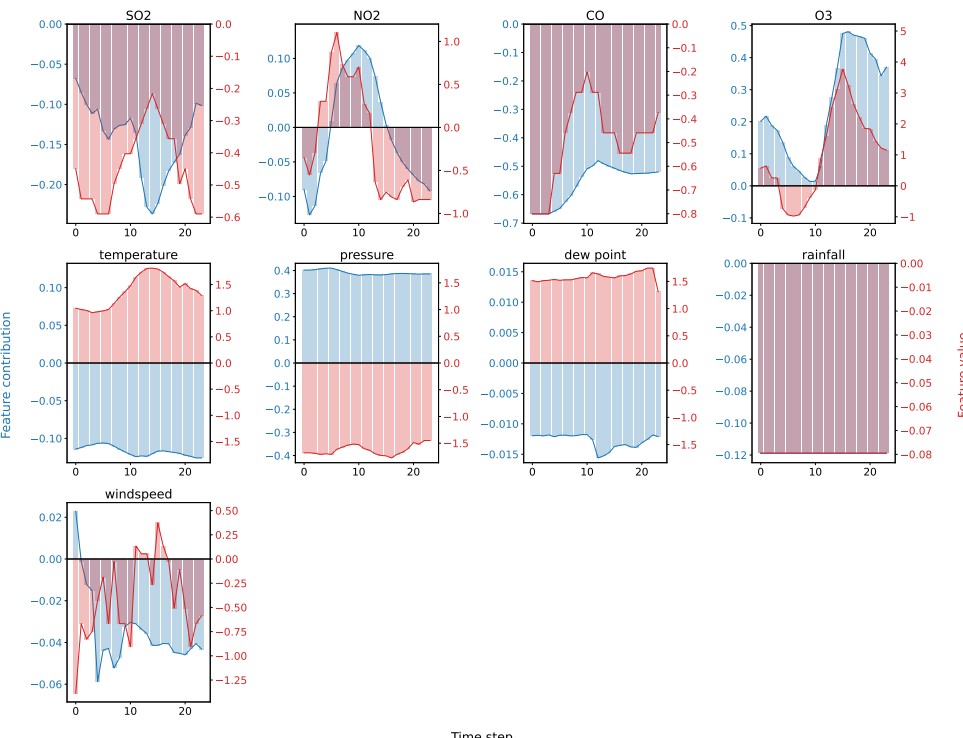

Figure 5: Local time-dependent feature contributions.

**Time-step importance:** We plot the average attention scores at the last time step $T$ in Figure 2. The process for extracting the average attention score of time step $u$ at time step $t$ is formalized as $\sum_{k=1}^{K} a_{k,t,u}$. This process is repeated over all data samples, and the results are averaged. Based on Figure 2, it seems that GATSM pays more attention to the initial and last states than to the intermediate states. This indicates that the current concentration of particulate matter depends on the initial state.

**Global feature contribution:** Figure 3 illustrates the global behavior of features in the AirQuality dataset, with red bars indicating the density of training samples. We extract $\sum_{k=1}^{K} h_b\left(x_{t,m}\right) w_{m,b}^{nbm} w_{k,m}^{out}$ from GATSM and repeat this process over the range of minimum to maximum feature values to plot the line. We found that the behavior of *SO2*, *O3*, and *windspeed* is inconsistent with prior human knowledge. Typically, high levels of *SO2* and *O3* are associated with poor air quality. However, GATSM learned that particulate matter concentration starts to decrease when *SO2* exceeds 10 and *O3* exceeds 5. This discrepancy may be due to sparse training samples in these regions, leading to insufficient training, or there may be interactions with other features. Another known fact is that high *windspeed* decreases particulate matter concentration. This is consistent when *windspeed* is below 0.7 in our observation. However, particulate matter concentration drastically increases when *windspeed* exceeds 0.7, likely due to the wind causing yellow dust.

**Local time-independent feature contribution:** To interpret the prediction of a data sample, we plot the local time-independent feature contributions, $\sum_{k=1}^{K} h_b\left(x_{t,m}\right) w_{m,b}^{nbm} w_{k,m}^{out}$, in Figure 4. The main x-axis (blue) represents feature contribution, the sub x-axis (red) represents feature value, and the y-axis represents time steps. We found that *SO2*, *NO2*, *CO*, and *O3* have positive correlations. In contrast, *temperature*, *pressure*, *dew point*, and *windspeed* have negative correlations. These are consistent with the global interpretations shown in Figure 3. Rainfall has the same values across all time steps.

**Local time-dependent feature contribution:** We also visualize the local time-dependent feature contributions, $\sum_{k=1}^{K} a_{k,t,u} h_b\left(x_{t,m}\right) w_{m,b}^{nbm} w_{k,m}^{out}$. Figure 5 illustrates the interpretation of the same data sample as in Figure 4. The time-dependent interpretation differs slightly from the time-independent interpretation. We found that there are time lags in *SO2*, *NO2*, *CO*, and *O3*, meaning previous feature values affect current feature contributions. For example, in the case of *SO2*, low feature values around time step 5 lead to low feature contributions around time step 13.

# 6 Future Works & Conclusion

Although GATSM achieved state-of-the-art performance within the transparent model category, it has several limitations. This section discusses these limitations and suggests future work to address them. GAMs have relatively slower computational times and larger model sizes compared to black-box models because they require the same number of feature functions as input features. To address this problem, methods such as the basis strategy can be proposed to reduce the number of feature functions, or entirely new methods for transparent models can be developed. The attention mechanism in GATSM may be a bottleneck. Fast attention mechanisms proposed in the literature [39, 40, 41, 42, 43], or the recently proposed Mamba [44], can help overcome this limitation. Existing time series models, including GATSM, only handle discrete time series and have limited length generalization ability, resulting in significantly reduced performance when very long sequences, unseen during training, are input. Extending GATSM to continuous models using NeuralODE [45] or HiPPO [46] could address this issue. GATSM still cannot learn higher-order feature interactions internally and shows low performance on complex datasets. Feature interaction methods proposed for transparent models may help address this problem [29, 15].

In this papre, we proposed a novel transparent model for time series named GATSM. GATSM consists of time-sharing NBM and the temporal module to effectively learn feature representations and temporal patterns while maintaining transparency. The experimental results demonstrated that GATSM has superior generalization ability and is the only transparent model with performance comparable to Transformer. We provided various visual interpretations of GATSM, demonstrated that GATSM capture interesting patterns in time series data. We anticipate that GATSM will be widely adopted in various fields and demonstrate strong performance. The broader impacts of GATSM across various fields can be found in Appendix A.

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

# A  Broader impact

We discuss the expected impacts of GATSM across various fields.

- **Time series adaptation:** GATSM extends existing GAMs to time series, enabling tasks that traditional GAMs could not perform in this context - e.g., better performance on time series and finding temporal patterns.

- **Improved decision-making system:** GATSM can show users their exact decision-making process, providing trust and confidence in its predictions to users. This enables decision-makers to make more informed choices, crucial in high-stakes domains such as healthcare.

- **Ethical AI:** GATSM can examine that their outcomes are biased or discriminatory by displaying the shape of feature functions. This is particularly important in ethically sensitive domains, such as recidivism prediction.

- **Scientific discovery:** Transparent models have already been used in various research fields for scientific discovery [47, 48]. GATSM also can be applied to these domains to obtain novel scientific insights.

Despite these advantages, it is important to remember that the interpretations of transparent models do not necessarily reflect exact causal relationships. While transparent models provide clear and faithful interpretations, they are still not capable of identifying causal relationships. Causal discovery is a complex task that requires further research.

# B  Dataset details

We use eight publicly available datasets for our experiments. Three datasets - Energy, Rainfall, and AirQuality - can be downloaded from the Monash repository [32]. Another three datasets - Heartbeat, LSST, and NATOPS - are available from the UCR repository [33]. The remaining two datasets can be downloaded from the PhysioNet [34]. Details of the datasets are provided below:

- **Energy** [49]: This dataset consists of 24 features related to temperature and humidity from sensors and weather conditions. These features are measured every 10 minutes. The goal of this dataset is to predict total energy usage.

- **Rainfall** [50]: This dataset consists of temperatures measured hourly. The goal of this dataset is to predict total daily rainfall in Australia.

- **AirQuality** [51]: This dataset consists of features related to air pollutants and meteorological data. The goal of this dataset is to predict the PM10 level in Beijing.

- **Heartbeat** [52]: This dataset consists of heart sounds collected from various locations on the body. Each sound was truncated to five seconds, and a spectrogram of each instance was created with a window size of 0.061 seconds with a 70% overlap. The goal of this dataset is to classify the sounds as either normal or abnormal.

- **Mortality** [53] This dataset consists of records of adult patients admitted to the ICU. The input features include the patient demographics, vital signs, and lab results. The goal of this dataset is to predict the in-hospital death of patients.

- **Sepsis** [54]: This dataset consists of records of ICU patients. The input features include patient demographics, vital signs, and lab results. The goal of this dataset is to predict sepsis six hours in advance at every time step.

- **LSST** [55]: This challenge dataset aims to classify astronomical time series. These time series consist of six different light curves, simulated based on the data expected from the Large Synoptic Survey Telescope (LSST).

- **NATOPS** [56]: This dataset aims to classify the Naval Air Training and Operating Procedures Standardization (NATOPS) motions used to control aircraft movements. It consists of 24 features representing the x, y, and z coordinates for each of the eight sensor locations attached to the body.

We used `get_UCR_data()` and `get_Monash_regression_data()` functions in the `tsai` library [57] to load the UCR and Monash datasets.

Table 6: Optimal hyper-parameters for GATSM.

**GATSM:** [256, 256, 128] hidden dims, 100 basis functions

| Dataset | Batch Size | NBM Batch Norm. | NBM Dropout | Attn. Embedding Size | Attn. Heads | Attn. Dropout | Learning Rate | Weight Decay |
|---|---|---|---|---|---|---|---|---|
| Energy | 32 | False | 2.315e-1 | 110 | 8 | 6.924e-2 | 4.950e-3 | 1.679e-3 |
| Rainfall | 32,768 | False | 5.936e-3 | 44 | 7 | 1.215e-3 | 9.225e-3 | 2.204e-6 |
| AirQuality | 4,096 | False | 2.340e-2 | 81 | 8 | 1.169e-1 | 6.076e-1 | 5.047e-6 |
| Heartbeat | 64 | True | 1.749e-1 | 92 | 2 | 1.653e-1 | 8.061e-3 | 4.787e-6 |
| Mortality | 512 | False | 7.151e-2 | 125 | 8 | 7.324e-1 | 7.304e-3 | 2.181e-4 |
| Sepsis | 512 | True | 6.523e-2 | 90 | 6 | 8.992e-1 | 4.509e-3 | 2.259e-2 |
| LSST | 1,024 | False | 2.500e-2 | 59 | 7 | 2.063e-1 | 5.561e-2 | 5.957e-3 |
| NATOPS | 64 | True | 4.827e-3 | 49 | 8 | 7.920e-1 | 8.156e-3 | 2.748e-2 |

## C   Implementation details

We use 13 models, including GATSM, for our experiments. We implement XGBoost and EBM using the `xgboost` [35] and `interpretml` [23] libraries, respectively. For NodeGAM, we employ the official implementation provided by its authors [8]. The remaining models are developed using PyTorch [36]. In addition, we implement the feature functions in NAM and NBM using grouped convolutions [58, 59] to enhance their efficiency. XGBoost and EBM are trained on two AMD EPYC 7513 CPUs, while the other models are trained on an NVIDIA A100 GPU with 80GB VRAM. All models undergo hyperparameter tuning via Optuna [37] with the Tree-structured Parzen Estimator (TPE) algorithm [60] in 100 trials. The hyperparameter search space and the optimal hyperparameters for the models are provided below:

• **XGBoost:** We tune the `n_estimators` in the integer interval [1, 1000], `max_depth` in the integer interval [0, 2000], learning rate in the continuous interval [1e-6, 1], `subsample` in the continuous interval [0, 1], and `colsample_bytree` in the continuous interval [0, 1].

• **MLP, NAM, NBM and NATM:** We tune the `batchnorm` in the descret set {False, True}, `dropout` in the continuous interval [0, 0.9], `learning_rate` in the continuous interval [1e-3, 1e-2], and `weight_decay` in the continuous interval [1e-6, 1e-1] on a log scale.

• **RNN, GRU and LSTM:** We tune the `hidden_size` in the integer interval [8, 128], `dropout` in the continuous interval [0, 0.9], `learning_rate` in the continuous interval [1e-3, 1e-2], and `weight_decay` in the continuous interval [1e-6, 1e-1] on a log scale.

• **Transformer:** We tune the `n_layers` in the integer interval [1, 4], `emb_size` in the integer interval [8, 32], `hidden_size` in the integer interval [8, 128], `n_heads` in the integer interval [1, 8], `dropout` in the continuous interval [0, 0.9], `learning_rate` in the continuous interval [1e-3, 1e-2], and `weight_decay` in the continuous interval [1e-6, 1e-1] on a log scale.

• **Linear:** We tune the `learning_rate` in the continuous interval [1e-3, 1e-2], and `weight_decay` in the continuous interval [1e-6, 1e-1] on a log scale.

• **EBM:** We tune `max_bins` in the integer interval [8, 512], `min_samples_leaf` and `max_leaves` in the integer interval [1, 50], `inner_bags` and `outer_bags` in the integer interval [1, 128], `learning_rate` in the continuous interval [1e-6, 100] on a log scale, and `max_rounds` in the integer interval [1000, 10000].

• **NodeGAM:** We tune `n_trees` in the integer interval [1, 256], `n_layers` and depth in the integer intervals [1, 4], `dropout` in the continuous interval [0, 0.9], `learning_rate` in the continuous interval [1e-3, 1e-2], and `weight_decay` in the continuous interval [1e-6, 1e-1] on a log scale.

• **GATSM:** We tune `nbm_batchnorm` in the descret set {False, True}, `nbm_dropout` in the continuous interval [0, 0.9], `attn_emb_size` in the integer interval [8, 128], `attn_n_heads` in the integer interval [1, 8], `attn_dropout` in the continuous interval [0, 0.9], `learning_rate` in the continuous interval [1e-3, 1e-2], and `weight_decay` in the continuous interval [1e-6, 1e-1] on a log scale. The optimal hyper-parameters for GATSM across all experimental datasets are provided in Table 6.

## D  Additional experiments

### D.1  Inference speed

The inference speed of machine learning models is a crucial metric for real-world systems. We evaluate the throughput of various models. The results are presented in Table 7. Since the datasets have fewer features than the number of basis functions in NBM, NAM achieves higher throughput than NBM. Transparent tabular models typically exhibit fast speeds. However, their throughput significantly decreases in datasets with many features, such as Heartbeat, Mortality, and Sepsis, because they require the same number of feature functions as the number of input features. Transformer shows higher throughput than the transparent time series models because it does not require feature functions, which are the main bottleneck of transparent models. Additionally, the PyTorch implementation of Transformer uses the flash attention mechanism [61] to enhance its efficiency. NATM has slightly higher throughput than GATSM, as it does not require the attention mechanism and has fewer feature functions compared to the number of basis functions in GATSM.

Table 7: Inference throughput of different models.

|  | Energy | Rainfall | AirQuality | Heartbeat | Mortality | Sepsis | LSST | NATOPS |
|---|---|---|---|---|---|---|---|---|
| NAM | 65.3K | 1.8M | 5.1M | 139.1K | 772.2K | 23.9K | 2.3M | 147.9K |
| NBM | 45.5K | 1.1M | 1.0M | 55.9K | 375.8K | 6.5K | 1.6M | 85.6K |
| Transformer | 30.9K | 240.5K | 174.2K | 15.7K | 161.9K | 134.6K | 214.4K | 68.3K |
| NATM | 5.3K | 699.3K | 241.3K | 1.3K | N/A | N/A | 28.6K | 19.2K |
| GATSM | 6.1K | 350.6K | 192.8K | 1.2K | 4.9K | 3.8K | 126.5K | 12.5K |

### D.2  Number of basis functions

We evaluate GATSM by varying the number of basis functions in the time-sharing NBM. The results for forecasting, binary classification, and multi-class classification datasets are presented in Figure 6. For the Sepsis dataset, using 200 and 300 basis functions causes the out-of-memory error. For the Energy and Heartbeat datasets, performance improves up to 100 basis functions but shows no further benefit when the number of bases exceeds 100. In other datasets, performance changes are not significant with different numbers of basis functions. In addition, there is a trade-off between the number of basis functions and computational speed. Therefore, we recommend generally setting the number of basis functions to 100. Note that the performance of GATSM with this hyper-parameter depends on the dataset size and complexity. Hence, a larger number of basis functions may benefit more complex datasets.

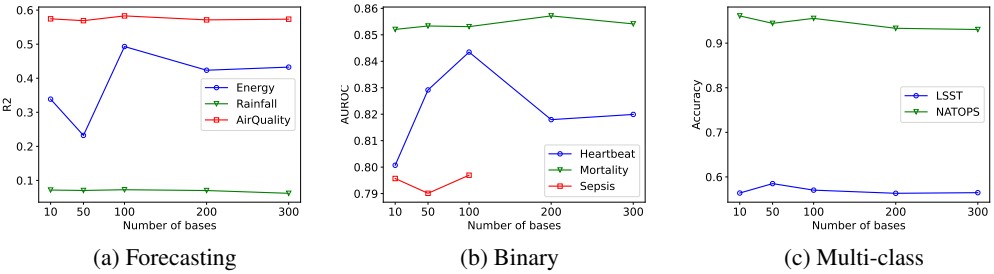

(a) Forecasting  (b) Binary  (c) Multi-class

Figure 6: Performances of GATSM on the different number of basis functions.

## E  Additional visualizations

In addition to the interpretations on the AirQuality dataset in section 5.4, we present another interesting interpretations of GATSM on the Rainfall dataset.

**Time-step importance:** Figure 7 illustrates the average importance of all time steps at the final time step. The importance exhibit a cyclical pattern of rising and falling at regular intervals, indicating that GATSM effectively captures seasonal patterns in the Rainfall dataset.

**Global feature contribution:** Figure 8 illustrates the global behavior of features in the Rainfall dataset, with red bars indicating the density of training samples. Our findings indicate that low *Max Temperature* and high *Min Temperature* contribute to an increase in rainfall.

**Local time-independent feature contribution:** Figure 9 shows the local time-independent feature contributions. Consistent with the global interpretation, *Avg. Temperature* and *Min Temperature* have positive correlations with rainfall, while *Max Temperature* has a negative correlation with rainfall.

**Local time-dependent feature contribution:** Figure 10 shows the local time-dependent feature contributions. All features exhibit patterns similar to the local time-independent contributions. However, we found that *Avg. Temperature* and *Min Temperature* have time lags between feature values and contributions.

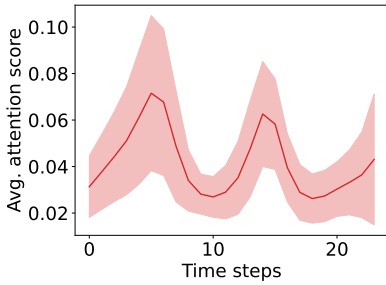

Figure 7: Average attention scores of time steps on the Rainfall dataset.

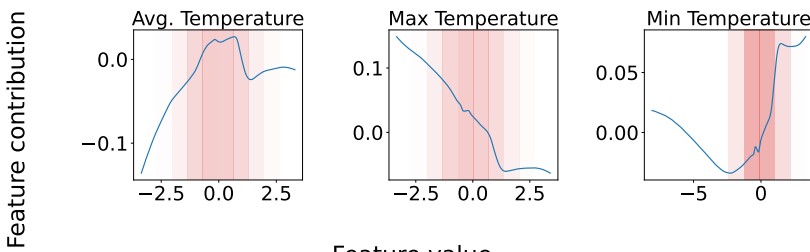

Figure 8: Global interpretations of features in the Rainfall dataset.

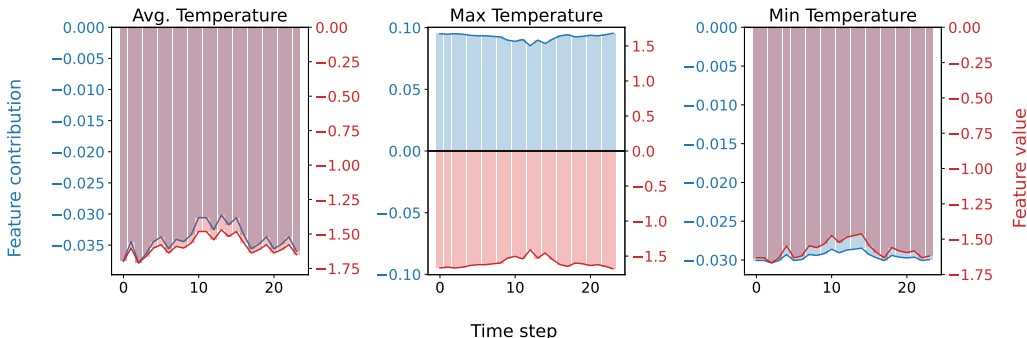

Figure 9: Local time-independent contributions of features in the Rainfall dataset.

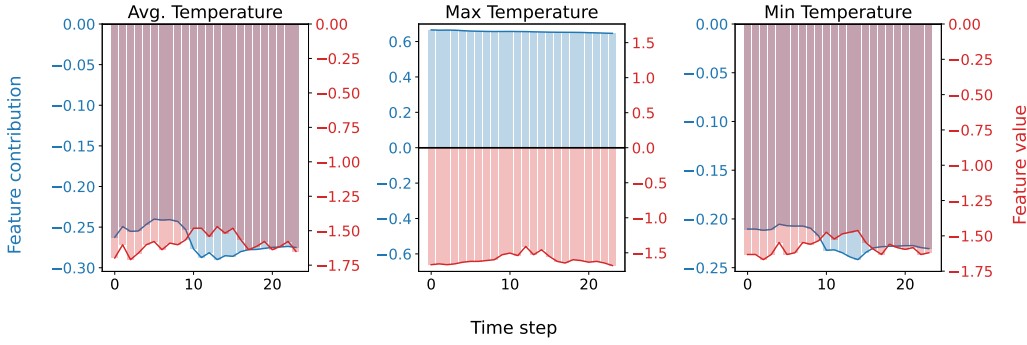

Figure 10: Local time-dependent contributions of features in the Rainfall dataset.

