# OpenReview forum: "Transparent Networks for Multivariate Time Series"
_NeurIPS.cc/2024/Conference — Submitted to NeurIPS 2024_

### Official Review · Reviewer_1uUQ · 2024-06-12

**Soundness:** 2
**Presentation:** 3
**Contribution:** 2
**Rating:** 5
**Confidence:** 4

**Summary:**

GATSM effectively captures temporal patterns and handles dynamic-length time series while preserving transparency, outperforming existing GAMs and matching the performance of black-box models like RNNs and Transformers.

**Strengths:**

* This paper is easy to understand.
* GATSM can be understood as a linear representation with good transparency.
* Surprisingly, this method improves performance while providing better interpretability compared with black-box.

**Weaknesses:**

* It is not clear how multi-head attention works in Definition 3.1 to learn temporal patterns. My understanding is that the global feature interacts with the current feature in Eq.3, so why is it that the input to the attention is not $x$ but the transformed $\tilde_x$. And then, are temporal patterns captured from attention?

* The addition of attention to NBM is under-motivated, so why not just replace $w^{nbm}$ to attention weight, so that you can learn one less set of parameters $w^{nbm}$.

* The experiment lacks DLinear [1], a strong baseline and providing interpretability. I think it's highly relevant.
1. Are Transformers Effective for Time Series Forecasting? AAAI 2023.

* Given the emphasis on interpretation or white-box modeling, qualitative experiments of the contributions/explanations need to be compared rather than visualization. If there is no ground-truth of the contribution, occlusion experiments in post-hoc methods [2, 3] can also be designed to explore the trade-off between performance and additive features.
2. Encoding time-series explanations through self-supervised model behavior consistency, NeurIPS 2023.
3. TimeX++: learning time-series explanations with information bottleneck, ICML 2024.

* In Table 7, why is the throughput of GATSM so much lower than NBM? Does NBM add up as features at the time level without sharing?

**Questions:**

See the weaknesses.

**Limitations:**

It's hard to deal with the higher-order interactions, e.g. GA^2M/GA^NM in time series, since time series often have long time points and the high complexity of gam-based techniques. In addition, the temporal-level causal interrelationships can be further explored.

---

> ### Author Rebuttal · Authors · 2024-08-06
>
> **Q1**: It is not clear how multi-head attention works in Definition 3.1 to learn temporal patterns. My understanding is that the global feature interacts with the current feature in Eq.3, so why is it that the input to the attention is not but the transformed \tilde_x. And then, are temporal patterns captured from attention?
>
> **A1**: In this paper, we aim to separately model the effects of features using NBM and temporal patterns using the attention mechanism to maintain transparency. In Equation (5), the input to the attention mechanism is $\textbf{v}$, which is a linearly transformed version of $\tilde{\textbf{x}}$ with positional encoding. In Equations (7) and (8), the current time step $\textbf{v}_i$ interacts with the previous time step $\textbf{v}_j$. Therefore, the attention mechanism can capture temporal patterns. In Equation (11), we rewrote the vector-level representation of the final prediction of GATSM in scalar form. This demonstrates that the prediction of GATSM satisfies Definition 3.1 by encapsulating the attention scores and NBM (i.e., $h_b$) into a function $f\_{u,m}$.
>
> **Q2**: The addition of attention to NBM is under-motivated, so why not just replace w^nbm to attention weight, so that you can learn one less set of parameters w^nbm.
>
> **A2**: $w^{nbm}$ and the attention score $\alpha$ have different roles. $w^{nbm}$ is applied to the basis functions in time-sharing NBM. Thus, it can be viewed as feature-wise weight. In contrast, the attention score $\alpha$ is applied to time-steps, serving as time-wise weight. Therefore, employing both $w^{nbm}$ and $\alpha$ improves the expressivity of GATSM.
>
> **Q3**: The experiment lacks DLinear [1], a strong baseline and providing interpretability. I think it's highly relevant.
>
> **A3**: We have added new experimental results comparing GATSM with two recent models, suggested DLinear and PatchTST. These experiments were conducted using three popular forecasting datasets—electricity, traffic, and weather—to evaluate 24-step, 48-step, and 72-step forecasting accuracy. We reported the average MAPEs and standard deviations over five random seed. The results show that PatchTST performs best in multi-step forecasting tasks, while GATSM demonstrates lower accuracy compared to recent state-of-the-art methods. However, GATSM has a unique advantage: it can provide fully faithful interpretations of the model. Additionally, GATSM is capable of handling dynamic time series, and we have reported its performance on various tasks, including forecasting, binary classification, and multi-class classification. We will add a figure illustrating the interpretations of GATSM and DLinear in multi-step forecasting
>
> ||Electricity|||Traffic|||Weather|||
> |---------------|--------------------|---------------|-------------------|---------------|-------------------|---------------|---------------|---------------|---------------|
> ||24h|48h|72h|24h|48h|72h|24h|48h|72h|
> |GATSM|0.122(+-0.005)|0.135(+-0.010)|0.137(+-0.007)|0.314(+-0.030)|0.302(+-0.046)|0.347(+-0.108)|0.859(+-0.115)|0.966(+-0.182)|0.935(+-0.133)|
> |PatchTST|0.100(+-0.005)|0.107(+-0.007)|0.104(+-0.005)|0.209(+-0.005)|0.227(+-0.013)|0.228(+-0.007)|0.622(+-0.006)|0.580(+-0.022)|0.581(+-0.019)|
> |Dlinear|0.108(+-0.003)|0.115(+-0.004)|0.113(+-0.001)|0.242(+-0.011)|0.250(+-0.007)|0.252(+-0.012)|0.844(+-0.085)|0.860(+-0.013)|0.817(+-0.032)|
>
> **Q4**: Given the emphasis on interpretation or white-box modeling, qualitative experiments of the contributions/explanations need to be compared rather than visualization. If there is no ground-truth of the contribution, occlusion experiments in post-hoc methods [2, 3] can also be designed to explore the trade-off between performance and additive features.
>
> **A4**: We conducted feature occlusion experiments using five datasets, each with more than ten input features. We determined the importance of features by averaging the absolute magnitude of their contributions and retrained GATSM without the bottom 20%, 40%, 60%, and 80% of features. The experimental results show a slight decrease in performance as the occlusion rate increases in the Mortality and Sepsis datasets, indicating that GATSM effectively identified important features. Additionally, performance increases as the occlusion rate increases in the Energy, Heartbeat, and NATOPS datasets, suggesting that GATSM can identify noisy features.
>
> ||occlusion rate||||
> |---------------|---------------|---------------|---------------|---------------|
> ||20%|40%|60%|80%|
> Energy ($\uparrow R^2$)|0.408(+-0.111)|0.487(+-0.062)|0.464(+-0.099)|0.562(+-0.098)|
> Heartbeat ($\uparrow AUROC$)|0.811(+-0.049)|0.838(+-0.028)|0.815(+-0.048)|0.806(+-0.085)|
> Mortality ($\uparrow AUROC$)|0.857(+-0.018)|0.853(+-0.018)|0.85(+-0.019)|0.846(+-0.016)|
> Sepsis ($\uparrow AUROC$)|0.796(+-0.005)|0.793(+-0.005)|0.784(+-0.013)|0.768(+-0.017)|
> NATOPS ($\uparrow Acc.$)|0.947(+-0.032)|0.953(+-0.027)|0.956(+-0.033)|0.961(+-0.025)|
>
> **Q5**: In Table 7, why is the throughput of GATSM so much lower than NBM? Does NBM add up as features at the time level without sharing?
>
> **A5**: The feature functions in NAM, NBM, and GATSM share learnable parameters across time steps (i.e., time-sharing). GATSM additionally requires the computation of attention scores, resulting in lower throughput compared to NBM.

---

> > ### Comment · Reviewer_1uUQ · 2024-08-10
> >
> > Thanks for the detailed response. My major concerns have been well addressed, and I have decided to raise my rating to 5.

---

### Official Review · Reviewer_f5AT · 2024-07-06

**Soundness:** 2
**Presentation:** 2
**Contribution:** 3
**Rating:** 5
**Confidence:** 4

**Summary:**

This paper introduces GATSM(Generalized Additive Time Series Model), designed for handling multivariate time series data with a focus on transparency and interpretability. Using independent networks to learn feature representations and transparent temporal modules to learn cross-time step dynamics, GATSM effectively learns temporal patterns and maintains interpretability.  It achieves comparable results achieves comparable performance to black-box time series models on various datasets, and proves the transparent predictions with cases.

**Strengths:**

1. A key strength of GATSM is its focus on transparency, providing clear insights into the decision-making process, which is crucial for applications in high-stakes domains like healthcare.
2. The paper presents a thorough evaluation across multiple datasets, including Energy, Rainfall, AirQuality, and several healthcare datasets, showcasing the model's robustness and generalization capabilities.
3. The authors provide extensive details about the experimental setup, including data splits, hyperparameters, and computational resources, facilitating reproducibility.

**Weaknesses:**

1.  The need for a large number of feature functions can limit scalability (even if it is reduced from TxM to B), particularly with high-dimensional data.
2. The dataset tasks encompass both 1-step forecasting and classification, each requiring distinct evaluation metrics. Presenting all results in a single table without proper clarification leads to confusion.
3. The selected baseline black-box models are relatively simple. Consider including one or two state-of-the-art methods for a more comprehensive comparison.

**Questions:**

Can this model be applied to a multi-step prediction explanation since one-step forecasting is simple to real-world applications.

**Limitations:**

Yes

---

> ### Author Rebuttal · Authors · 2024-08-06
>
> **Q1**: The need for a large number of feature functions can limit scalability (even if it is reduced from TxM to B), particularly with high-dimensional data.
>
> **A1**: A large number of feature functions in GAM are required to maintain its transparency, which is a key limitation. We will address this limitation in future work.
>
> **Q2**: The dataset tasks encompass both 1-step forecasting and classification, each requiring distinct evaluation metrics. Presenting all results in a single table without proper clarification leads to confusion.
>
> **A2**: We agree with your comment. We will add the names of the evaluation metrics to the table and provide a description of both the table and the metrics in the caption.
>
> **Q3**: The selected baseline black-box models are relatively simple. Consider including one or two state-of-the-art methods for a more comprehensive comparison.
>
> **A3**: We have added new experimental results comparing GATSM with two recent models, PatchTST and DLinear. These experiments were conducted using three popular forecasting datasets—electricity, traffic, and weather—to evaluate 24-step, 48-step, and 72-step forecasting accuracy. We reported the average MAPEs and standard deviations over five random seed. The results show that PatchTST performs best in multi-step forecasting tasks, while GATSM demonstrates lower accuracy compared to recent state-of-the-art methods. However, GATSM has a unique advantage: it can provide fully faithful interpretations of the model. Additionally, GATSM is capable of handling dynamic time series, and we have reported its performance on various tasks, including forecasting, binary classification, and multi-class classification.
>
> ||Electricity|||Traffic|||Weather|||
> |---------------|--------------------|---------------|-------------------|---------------|-------------------|---------------|---------------|---------------|---------------|
> ||24h|48h|72h|24h|48h|72h|24h|48h|72h|
> |GATSM|0.122(+-0.005)|0.135(+-0.010)|0.137(+-0.007)|0.314(+-0.030)|0.302(+-0.046)|0.347(+-0.108)|0.859(+-0.115)|0.966(+-0.182)|0.935(+-0.133)|
> |PatchTST|0.100(+-0.005)|0.107(+-0.007)|0.104(+-0.005)|0.209(+-0.005)|0.227(+-0.013)|0.228(+-0.007)|0.622(+-0.006)|0.580(+-0.022)|0.581(+-0.019)|
> |Dlinear|0.108(+-0.003)|0.115(+-0.004)|0.113(+-0.001)|0.242(+-0.011)|0.250(+-0.007)|0.252(+-0.012)|0.844(+-0.085)|0.860(+-0.013)|0.817(+-0.032)|
>
> **Q4**: Can this model be applied to a multi-step prediction explanation since one-step forecasting is simple to real-world applications.
>
> **A4**: Our GATSM can be applied to multi-step forecasting tasks and can provide interpretations for these tasks. We have reported GATSM's performance on multi-step forecasting datasets above. Additionally, we will add a figure illustrating the interpretation of GATSM in multi-step forecasting.

---

> > ### Comment · Reviewer_f5AT · 2024-08-11
> >
> > Thanks for the detailed response. My major concerns have been well addressed, and I believe they will be addressed in the final versions thus I decided to raise my rating to 5.

---

### Official Review · Reviewer_cPMr · 2024-07-12

**Soundness:** 3
**Presentation:** 3
**Contribution:** 2
**Rating:** 6
**Confidence:** 1

**Summary:**

The paper introduces a Generalized Additive Model for time series, combining feature embedding and attention layer. The proposed solution is evaluated on forecasting, binary and multiclass classification, over 8 datasets, against black box and transparent models. Global, local, time-focused and feature focused interpretability methods are provided.

**Strengths:**

Experimental evaluation is convincing: the presented model has state of the art performance. On the transparency side, the comparison between models allows the author to postulate on the presence of absence of interactions of covariates, and time-specific patterns. Evaluations of weights at the final linear layer give varied visualizations.

The reasoning behind the model construction and component is clear, and ablation experiment for each component were provided.

Paper is clear with no major problem in writing.

**Weaknesses:**

I am not sure if the work is original. The idea of using DNN first on the time axis without covariate interaction is not new, but wether there is a model similar to the proposed solution, I do not know. The review of previous works focuses on Generalized Additive Models, but a similar neural structure may have been presented without being positioned as a GAM.

One improvement to be done would be to add more clarity on figure captions. In its present form, it requires back and forth to the text to understand both what is plotted and what conclusions to draw from it.

**Questions:**

In forecasting tasks, there are often long horizons to be considered. Is it possible to build a multioutput model for forecasting using GAM formalism? In which case, would the interpretation be able to relate future timestamps together, as series are often autocorrelated?

Can correlation among dimensions be detected with GAM? This would be especially relevant in high dimensional MTS.

**Limitations:**

Several limitations are identified: possible overfitting of the model due to overparameterization in NBM part, slow attention mechanisms that do not benefit from state of the art methods, and the fact that it was not evaluated for long sequences (and might not be suited for them).

---

> ### Author Rebuttal · Authors · 2024-08-05
>
> **Q1**: I am not sure if the work is original. The idea of using DNN first on the time axis without covariate interaction is not new, but wether there is a model similar to the proposed solution, I do not know. The review of previous works focuses on Generalized Additive Models, but a similar neural structure may have been presented without being positioned as a GAM.
>
> **A1**: Several methods, such as DeepAR, apply neural networks along the time axis without covariate interaction, but they involve non-linear interactions with the current and previous states. As a result, they lack transparency, which is the main advantage of our GATSM. To the best of our knowledge, no transparent time series models have been proposed in the previous literature. We will add relevant time series models to the related work section.
>
> **Q2**: One improvement to be done would be to add more clarity on figure captions. In its present form, it requires back and forth to the text to understand both what is plotted and what conclusions to draw from it.
>
> **A2**: We will add detailed captions to the figures and tables to enhance their clarity and improve understanding.
>
> **Q3**: In forecasting tasks, there are often long horizons to be considered. Is it possible to build a multioutput model for forecasting using GAM formalism? In which case, would the interpretation be able to relate future timestamps together, as series are often autocorrelated?
>
> **A3**: Implementing multi-output and multi-step prediction systems using the GAM formalism is possible. We have conducted 24-step, 48-step and 72-step prediction experiments using GATSM and two recent time series forecasting models, PatchTST and DLinear on three population forecasting datasets, Electricity, Traffic and Weather. We reported the average MAPEs and standard deviations over five random seed. We believe that GATSM can capture autocorrelations between time steps in the auto-regressive setting because it learns temporal patterns across different time steps. If GATSM captures autocorrelation, the interpretation will relate different time steps together.
>
> ||Electricity|||Traffic|||Weather|||
> |---------------|--------------------|---------------|-------------------|---------------|-------------------|---------------|---------------|---------------|---------------|
> ||24h|48h|72h|24h|48h|72h|24h|48h|72h|
> |GATSM|0.122(+-0.005)|0.135(+-0.010)|0.137(+-0.007)|0.314(+-0.030)|0.302(+-0.046)|0.347(+-0.108)|0.859(+-0.115)|0.966(+-0.182)|0.935(+-0.133)|
> |PatchTST|0.100(+-0.005)|0.107(+-0.007)|0.104(+-0.005)|0.209(+-0.005)|0.227(+-0.013)|0.228(+-0.007)|0.622(+-0.006)|0.580(+-0.022)|0.581(+-0.019)|
> |Dlinear|0.108(+-0.003)|0.115(+-0.004)|0.113(+-0.001)|0.242(+-0.011)|0.250(+-0.007)|0.252(+-0.012)|0.844(+-0.085)|0.860(+-0.013)|0.817(+-0.032)|
>
> **Q4**: Can correlation among dimensions be detected with GAM? This would be especially relevant in high dimensional MTS.
>
> **A4**: GAM can capture correlations among different dimensions (or features). One straightforward approach is to input high-order features directly into GAM. For example, given three features $x_1$, $x_2$, and $x_3$, we can manually craft second-order features such as $x_1 \times x_2$, $x_2 \times x_3$, and $x_1 \times x_3$ and include them in the input to GAM. This allows GAM to learn second-order interactions, capturing correlations between pairs of dimensions. Additionally, recent methods for high-order interactions in transparent models, such as Scalable Polynomial Additive Model and High-order Neural Additive Model, can be employed. Using these methods enables GAM to capture correlations between different dimensions without manually crafted features.

---

> > ### Comment · Reviewer_cPMr · 2024-08-11
> >
> > Thank you for the answer. I believe that the answer to Q3 especially would improve the paper, paired with interpretation plots attending temporal patterns in multioutput models. I will pass my rating to weak accept, but I will keep my review confidence as 1, as proposing new neural structure isn't my domain.

---

### Official Review · Reviewer_JRxn · 2024-07-12

**Soundness:** 2
**Presentation:** 3
**Contribution:** 2
**Rating:** 4
**Confidence:** 2

**Summary:**

This work aims to build transparent models for the time series domain for better interpretability. Specifically, they proposed a work called Generalized Additive Time Series Model (GATSM) that consists of independent feature networks as well as a temporal attention module to learn temporal patterns. The corresponding model can be written into a scalar form to ensure interpretability/transparency. The authors applied their model to several datasets, and showed that GATSM can outperform existing generalized additive models. The model can also be used to interpret the features in original dataset.

**Strengths:**

- Important motivation. Building transparent models for time series is a critical task.
- The scalar representation of features seems clean (eq 11)

**Weaknesses:**

While this work is not of my direct expertise, I think the following contents have room for improvement:

1. Experimental results are weak.
- Black-box Time Series Models seem out-of-dated. The authors should consider better models such as TimesNet, PatchTST, FreqTransformer, or Informer for commonly used black-box models.
- Forecasting tasks use R2 score for evaluation, but an R2 score of 0.07 (or in general, below 0.5) seems very low. The authors should show some visualization examples to ensure the model is functioning.
- Figure 4/5 are not self-explainable, the authors should try to explain what is happening in those figures, and how the interpretability is quantified.
- The work could benefit from synthetic dataset, where casual relationships are manually crafted and thus can be evaluated.

**Questions:**

- Line 123: How one should understand the proposed basis functions, are they similar to quantized vectors in works like VQVAE?
- Eq 5: How would positional embeddings affect the interpretability of the model?
- How does the interpretability results differ in datasets with very long length (e.g. Mortality) v.s. that of shorter length?

**Limitations:**

Yes

---

> ### Author Rebuttal · Authors · 2024-08-05
>
> **Q1**: Black-box Time Series Models seem out-of-dated. The authors should consider better models such as TimesNet, PatchTST, FreqTransformer, or Informer for commonly used black-box models.
>
> **A1**: We have added new experimental results comparing GATSM with two recent models, PatchTST and DLinear. These experiments were conducted using three new datasets—electricity, traffic, and weather—to evaluate forecasting accuracy for 24-hour, 48-hour, and 72-hour predictions. We reported the average MAPEs and standard deviations over five random seed. The results show that PatchTST performs best in multi-step-ahead forecasting tasks, while GATSM demonstrates lower accuracy compared to recent state-of-the-art methods. However, GATSM has a unique advantage: it can provide fully faithful interpretations of the model. Additionally, GATSM is capable of handling dynamic time series, and we have reported its performance on various tasks, including forecasting, binary classification, and multi-class classification.
>
> ||Electricity|||Traffic|||Weather|||
> |---------------|--------------------|---------------|-------------------|---------------|-------------------|---------------|---------------|---------------|---------------|
> ||24h|48h|72h|24h|48h|72h|24h|48h|72h|
> |GATSM|0.122(+-0.005)|0.135(+-0.010)|0.137(+-0.007)|0.314(+-0.030)|0.302(+-0.046)|0.347(+-0.108)|0.859(+-0.115)|0.966(+-0.182)|0.935(+-0.133)|
> |PatchTST|0.100(+-0.005)|0.107(+-0.007)|0.104(+-0.005)|0.209(+-0.005)|0.227(+-0.013)|0.228(+-0.007)|0.622(+-0.006)|0.580(+-0.022)|0.581(+-0.019)|
> |Dlinear|0.108(+-0.003)|0.115(+-0.004)|0.113(+-0.001)|0.242(+-0.011)|0.250(+-0.007)|0.252(+-0.012)|0.844(+-0.085)|0.860(+-0.013)|0.817(+-0.032)|
>
> **Q2**: Forecasting tasks use R2 score for evaluation, but an R2 score of 0.07 (or in general, below 0.5) seems very low. The authors should show some visualization examples to ensure the model is functioning.
>
> **A2**: The scores of all models on the Rainfall dataset are low in the experiments, suggesting that this dataset is very complex and hard to forecast. Nevertheless, time series models significantly outperform tabular models, indicating that they effectively capture temporal patterns. We will add figures of forecasting results of GATSM on the Rainfall dataset to demonstrate that it is functioning appropriately.
>
> **Q3**: Figure 4/5 are not self-explainable, the authors should try to explain what is happening in those figures, and how the interpretability is quantified.
>
> **A3**: In Figures 4 and 5, the x-axis (left) represents feature contributions, the sub-x-axis (right) represents feature values, and the y-axis represents time steps. These visualizations illustrate the patterns between feature values and their contributions. Due to GATSM's separate modeling scheme of feature contribution and time importance, we can obtain both time-independent and time-dependent feature contributions. Figure 4 shows the time-independent feature contributions, indicating the effects of features if no temporal patterns exist. In contrast, Figure 5 shows the time-dependent feature contributions, indicating the effects of features while considering previous history. Therefore, the feature contributions in Figures 4 and 5 can differ. For example, in Figure 4, SO2, NO2, and CO show only a positive correlation between contribution and feature value. However, in Figure 5, time lags appear in these three features. We will add a more detailed description of Figures 4 and 5 in the manuscript.
>
> **Q4**: The work could benefit from synthetic dataset, where casual relationships are manually crafted and thus can be evaluated.
>
> **A4**: We believe that causal discovery and making precise predictions by taking into account confoundings using transparent models are promising directions for our future work. Since causality is outside the scope of this paper, we plan to pursue these works in future studies.
>
> **Q5**: Line 123: How one should understand the proposed basis functions, are they similar to quantized vectors in works like VQVAE?
>
> **A5**: Vector quantization and basis functions are similar concepts with slight differences. Both methods aim to decompose a function into smaller components or construct a function by combining multiple functions. However, the key difference is that vector quantization maps a continuous vector into a discrete space and uses a new vector corresponding to that mapped space. In contrast, basis functions construct a larger continuous function using multiple smaller continuous functions.
>
> **Q6**: Eq 5: How would positional embeddings affect the interpretability of the model?
>
> **A6**: Positional encodings do not affect the feature functions (NBM); they only influence the attention scores. This design choice is intended to separately model feature effects through NBM and temporal patterns through the attention mechanism, ensuring transparency in time series. Therefore, positional encoding does not decrease the interpretability of GATSM.
>
> **Q7**: How does the interpretability results differ in datasets with very long length (e.g. Mortality) v.s. that of shorter length?
>
> **A7**: Due to the transparency of GATSM, it consistently produces fully faithful interpretations of the model. As a result, the length of the time series does not affect the interpretability of GATSM. We will add figures similar to Figures 2, 3, 4, and 5 for long-length datasets.

---

> ### Comment · Reviewer_JRxn · 2024-08-12
> **Thanks**
>
> Thank you for the response. The new results show that GATSM performs worse on SOTA black-box networks on new datasets, and the authors did not provide results showing the performance of SOTA black-box networks on datasets presented in the paper. Additionally, the authors did not provide new synthetic results as I requested. Thus, I'd keep my score to be the same.

---

> ### Author Response · Authors · 2024-08-13
> **Response to Reviewer JRxn**
>
> Thank you for your feedback. We agree that the experiments you suggested will improve the quality of our paper. Thus, we have added the following new experimental results:
> - In the multi-step forecasting experiments, our PatchTST and DLinear outperform our GATSM. However, our primary contribution is the development of a novel, interpretable model for multivariate time series, GATSM, which offers human-understandable interpretations with a slight trade-off in accuracy.
> - We trained PatchTST and DLinear on the Energy, Rainfall, and AirQuality datasets, reporting their $R^2$ scores and standard deviations across five random seeds. However, performance results for the remaining five datasets are unavailable for the following reasons: the Mortality and Sepsis datasets have dynamic-length sequences, which are incompatible with forecasting models that require fixed-length observations. Additionally, the Heartbeat, LSST, and NATOPS datasets involve many-to-one classification tasks (binary or multi-class), a setting not supported by the publicly available implementations of PatchTST and DLinear.
>
> ||Energy($\uparrow R^2$)|Rainfall($\uparrow R^2$)|AirQuality($\uparrow R^2$)|
> |---------------|--------------------|---------------|-------------------|
> |DLinear|0.486(+-0.102)|0.086(+-0.014)|0.685(+-0.024)|
> |PatchTST|0.413(+-0.236)|0.098(+-0.019)|0.638(+-0.045)|
> |GATSM|0.493(+-0.173)|0.073(+-0.027)|0.583(+-0.026)|
>
> - We conducted an experiment using a synthetic tumor growth dataset[1] that simulates changes in tumor size over time. The tumor size at any given time step is influenced by three factors: prior tumor size, chemotherapy, and radiotherapy. We set the chemotherapy coefficient to 10, that is chemotherapy significantly reduce tumor. The experimental results showed that while GATSM underperforms compared to more complex black-box models, it still achieves strong accuracy ($R^2$ > 0.9). Additionally, we visualized the contributions of the three factors, revealing that recent time steps are notably more influential than earlier time steps due to the direct impact of prior tumor size. GATSM also effectively captured the effects of chemotherapy and radiotherapy on reducing tumor size, with chemotherapy having a significantly greater impact than radiotherapy.
>
> ||SyntheticTumorGrowth ($\uparrow R^2$)|
> |-|-|
> |Transformer|0.965(+-0.002)|
> |DLinear|0.956(+-0.009)|
> |PatchTST|0.953(+-0.006)|
> |GATSM|0.906(+-0.016)|
>
> [1] Geng, C., Paganetti, H., & Grassberger, C. (2017). Prediction of treatment response for combined chemo-and radiation therapy for non-small cell lung cancer patients using a bio-mathematical model. Scientific reports, 7(1), 13542.

---

### Author Rebuttal · Authors · 2024-08-06

### **Response to all reviewers**
We appreciate all the reviewers for their helpful comments and discussion on our manuscript. The feedback provided was instrumental in improving the quality of the manuscript. We have addressed each of the reviewers' questions and concerns individually.

---

### Decision · Program_Chairs · 2024-09-25

**Decision:**

Reject

**Comment:**

This paper proposes a Generalized Additive model for transparent and interpretable modeling of time series data.
The reviewers generally agree about the importance of the motivation, and the breadth of the datasets used in the empirical validation.
However, several concerns have been raised about the originality of the work, the clarity of the presentation, and the limitations of the model.
Specifically, reviewers expressed some doubts regarding the novelty of the approach and how transparency is being quantified concretely. These doubts are compounded by the sometime unclear exposition, in particular regarding the evaluation metrics and the presentations of the empirical results that fail to elucidate and quantify the gains in interpretability.
In addition, the reviewers have raised concerns about the benchmarking against black-box baselines, as the number of state-of-the-art methods that have been compared against the proposed method has been indicated as limited. The addition of new experimental results in the rebuttal is appreciated, but it is still not capturing the full spectrum of the state-of-the-art baselines currently available for time-series data, including more recent transformer-based architectures and foundation models.
Overall, the reviewers pointed out several strength of the paper, but not enough to counterbalance the concerns that were raised.